# Integrative Analysis of Metabolomic and Transcriptomic Data Reveals the Mechanism of Color Formation in Corms of *Pinellia ternata*

**DOI:** 10.3390/ijms24097990

**Published:** 2023-04-28

**Authors:** Rong Xu, Ming Luo, Jiawei Xu, Mingxing Wang, Bisheng Huang, Yuhuan Miao, Dahui Liu

**Affiliations:** Key Laboratory of Traditional Chinese Medicine Resources and Chemistry of Hubei Province, Hubei University of Chinese Medicine, Wuhan 430065, China; xurong890119@hotmail.com (R.X.); 3314@hbtcm.edu.cn (M.L.); 13871883140@163.com (J.X.); wmx1025182472@163.com (M.W.); hbsh1963@163.com (B.H.)

**Keywords:** *Pinellia ternata*, metabolome, transcriptome, epidermis color, anthocyanin

## Abstract

*Pinellia ternata* (Thunb.) Breit. (*P. ternata*) is a very important plant that is commonly used in traditional Chinese medicine. Its corms can be used as medicine and function to alleviate cough, headache, and phlegm. The epidermis of *P. ternata* corms is often light yellow to yellow in color; however, within the range of *P. ternata* found in JingZhou City in Hubei Province, China, there is a form of *P. ternata* in which the epidermis of the corm is red. We found that the total flavonoid content of red *P. ternata* corms is significantly higher than that of yellow *P. ternata* corms. The objective of this study was to understand the molecular mechanisms behind the difference in epidermal color between the two forms of *P. ternata*. The results showed that a high content of anthocyanidin was responsible for the red epidermal color in *P. ternata*, and 15 metabolites, including cyanidin-3-O-rutinoside-5-O-glucoside, cyanidin-3-O-glucoside, and cyanidin-3-O-rutinoside, were screened as potential color markers in *P. ternata* through metabolomic analysis. Based on an analysis of the transcriptome, seven genes, including *PtCHS1*, *PtCHS2*, *PtCHI1*, *PtDFR5*, *PtANS*, *PtUPD-GT2*, and *PtUPD-GT3*, were found to have important effects on the biosynthesis of anthocyanins in the *P. ternata* corm epidermis. Furthermore, two transcription factors (TFs), *bHLH1* and *bHLH2*, may have regulatory functions in the biosynthesis of anthocyanins in red *P. ternata* corms. Using an integrative analysis of the metabolomic and transcriptomic data, we identified five genes, *PtCHI*, *PtDFR2*, *PtUPD-GT1*, *PtUPD-GT2*, and *PtUPD-GT3*, that may play important roles in the presence of the red epidermis color in *P. ternata* corms.

## 1. Introduction

*Pinellia ternata* (Thunb.) Breit., known as Banxia in Chinese, is a commonly used natural medicinal plant. The corm of *P. ternata* is the main medicinal part, and it has been widely used in China, Japan, Republic of Korea, and other countries for more than 2000 years to treat coughs, expectoration, headaches, and other symptoms. Among the 588 traditional Chinese medicine (TCM) prescriptions with the highest frequency of issues in China, *P. ternata* currently ranks 22nd. Among the top 210 TCM prescriptions issued in Japan, *P. ternata* was used in 46 prescriptions, accounting for 22%. There is a catalog of 100 ancient classic prescriptions (the first batch) issued by the National Administration of Traditional Chinese Medicine of China, of which 17 prescriptions use *P. ternata*. The earliest use of the *P. ternata* corm as a medicinal material can be traced back to the Han Dynasty in preparations such as the “Xuanfu Daizhe Decoction”, “Bamboo leaf gypsum Decoction”, and “Banxia Xiexin Decoction”, in Zhang Zhongjing’s Febrile Diseases. In the present battle against COVID-19, both the Qingfei Paidu Decoction [1,2] and Huashi Baidu Granule [3,4], which have significant curative effects against COVID-19 infection, contain *P. ternata*. Thus, it is apparent that *P. ternata*, as a medicinal plant, is of great significance to human health.

It is well known that the color of the epidermis in the fresh corms of *P. ternata* is mainly light yellow to yellow, but we have found a natural variant of *P. ternata* in which the corm epidermis is red to red-purple in color. This variant was found in Jingzhou City, Hubei Province, China, which is the main area of *P. ternata* production. According to the results of previous studies, the color of plants is determined by their metabolites [5]. Flavonoids [6], carotenoids [7,8], anthocyanins [9,10,11], and other metabolites are the key factors that determine plant color. For example, the color of apricot fruit flesh is related to the content of carotenes [12]. Fruits with a high carotene content are yellow, whereas fruits with a low carotene content are white. Moreover, the nutritional value of yellow apricot fruit is higher than that of white apricot fruit. The content of tanshinone determines the color of the root epidermis in *Salvia miltiorrhiza* [13]. The *S. miltiorrhiza* root epidermis with a high tanshinone content is red, and the contents of its medicinal components, such as diterpenoids and dihydrotanshinone, are significantly higher than in *S. miltiorrhiza*, in which the root epidermis is white [13]. The anthocyanidin content of red-flesh apples is higher than that of non-red-flesh apples, and the contents of phenolic acids, total flavonoids, and other metabolites are also higher. In addition, red-flesh apples have demonstrated stronger tolerances to salt and drought than non-red-flesh apples [14]. *Colocasia esculenta* (taro), along with *P. ternata*, belongs to the Araceae family and may have similar metabolites. Taro corms can be either purple or white in color, and the anthocyanidin content plays a key role in determining the purple color in taro corms [15]. It can be observed that color is not only a plant appearance trait, but is also one of the manifestations of its internal quality. Flavonoids, as an important component of the pharmacological components of *P. ternata* [16,17], have antioxidant, anticancer, antihyperlipidemic, antiallergic, antibacterial, antiviral, and anti-inflammatory pharmacological activities. The biosynthesis of anthocyanins is an important component in the flavonoid metabolism pathway, and the formation of red on the epidermis of *P. ternata* may be an important epigenetic characteristic of changes in the content of flavonoids in *P. ternata* corms. Therefore, studying the mechanism of color formation in *P. ternata* is conducive to improving their quality.

Anthocyanins are a class of water-soluble pigments that are widely found in plant flowers, fruits, leaves, stems, and other tissues [18,19,20]. Anthocyanins are flavonoids and are the major colored substances responsible for the red to purple to blue coloration in plants. There are six types of anthocyanidins commonly found in plants; these include pelargonidin, cyanidin, delphinidin, peonidin, petunidin, and malvidin. Usually, delphinidin, petunidin, and malvidin contribute the purple-blue color to plants, pelargonidin imparts orange to red, and cyanidin and peonidin impart purple/red colors [5]. Anthocyanins not only play an important role in plant pollination and seed dispersal, but also help plants resist biotic and abiotic stresses. In plants, the pathway for anthocyanin biosynthesis starts from the precursor phenylalanine. This is catalyzed by a series of structural enzymes, including chalcone synthase (CHS), chalcone isomerase (CHI), favonoid-3-hydroxylase (F3H), favonoid-3′-hydroxylase (F3′H), favonoid-3′,5′-hydroxylase (F3′5′H), dihydrofavonol 4-reductase (DFR), and anthocyanidin synthase (ANS) to form unstable anthocyanidins. Finally, the unstable anthocyanidins are modified by glycosyltransferases (UPD-GTs) and methyltransferases (MT) to form stable anthocyanins [21,22,23]. In addition to the above enzyme-coding structural genes, anthocyanin biosynthesis is also regulated by WD40, bHLH, and R2R3-MYB transcription factors [24,25]. The anthocyanin biosynthesis pathway is well documented in many plants, but little is known about anthocyanin synthesis in *P. ternata*.

We measured the total flavonoid content of two different colored *P. ternata* corms. The total flavonoid content of red *P. ternata* corms was significantly higher than that of yellow *P. ternata* corms. To study the molecular and metabolic mechanism of color formation in *P. ternata*, we used high-performance liquid chromatography (HPLC) to determine whether the anthocyanidin content in the epidermis of red-corm *P. ternata* was significantly higher than that in yellow-corm *P. ternata*. We speculated that the higher anthocyanidin content was the reason for the formation of a red epidermis in some corms of *P. ternata*. The differentially expressed metabolites of red-corm *P. ternata* and yellow-corm *P. ternata* were analyzed using metabolomics, and 15 metabolites were identified as potential marker metabolites for the red corm form of *P. ternata*. We also sequenced the full-length transcriptomes of the flesh and epidermis of the red and yellow corms of *P. ternata*. By integrating transcriptomic and metabolomic analyses, we were able to infer the mechanism behind the formation of the red epidermis in *P. ternata* corms, and we identified some candidate genes related to the synthesis and regulation of anthocyanins; *PtDFR2*, *PtUPD-GT1*, *PtUPD-GT2*, *PtUPD-GT3*, and other genes may play key roles in anthocyanin biosynthesis in the *P. ternata* corm epidermis. This study provides a theoretical basis for improving the appearance and internal quality of *P. ternata.*

## 2. Results

### 2.1. Comparison of the Total Flavonoid Content of Two Different Colored P. ternata Corms and Differentially Accumulated Metabolites (DAMs) between the Flesh and Epidermis in the Red and Yellow Corms of P. ternata

In this study, we measured the content of total flavonoids, an important pharmacological component of *P. ternata*. The results showed that the total flavonoid content of the corms with a red epidermis (R, 44.9877 mg/g) was significantly higher than that of the corms with a yellow epidermis (Y, 28.0577 mg/g) (Figure 1C,D). We found that the red color mainly developed on the epidermis of *P. ternata* corms (Figure 1A,B). In order to investigate the differences in metabolites between red and yellow *P. ternata* corms, the epidermis and flesh were collected from corms of both colors, and four different samples were obtained: Yf (flesh of yellow-corm *P. ternata*), Ye (epidermis of yellow-corm *P. ternata*), Rf (flesh of red-corm *P. ternata*), and Re (epidermis of red-corm *P. ternata*). In our study, the anthocyanidin contents in the Yf, Ye, Rf, and Re samples were measured by HPLC (Appendix A). We found that only two types of anthocyanidin: pelargonidin and cyanidin, were detected in *P. ternata*. The contents of cyanidin in the two epidermal samples, Re (0.2933 mg/g) and Ye (0.1047 mg/g), were significantly higher than in the flesh samples, Rf (0.0289 mg/g) and Yf (0.0281 mg/g), respectively. In addition, the content of cyanidin in Re was significantly higher than that in Ye, but there was no difference in the content of cyanidin between Rf and Yf. The content of pelargonidin in Re (0.0457 mg/g) was significantly higher than that in Rf (0.0052 mg/g) and Ye (0.0073 mg/g), but there was no significant difference in the content of pelargonidin between Rf and Yf (0.0043 mg/g). The sum of the contents of cyanidin and pelargonidin can be regarded as the total anthocyanidin content in *P. ternata*. The total anthocyanidin contents of Yf (0.0323 mg/g), Rf (0.0341 mg/g), Ye (0.1121 mg/g), and Re (0.3390 mg/g) correlated with the cyanidin contents in the different samples (Figure 1D). Therefore, it can be inferred that the difference in the epidermal color between the red and yellow *P. ternata* corms is caused by differences in the contents of two types of anthocyanidin: cyanidin and pelargonidin.

As anthocyanidins represent one of the main branches of the flavonoid biosynthesis pathway, we determined the components and contents of flavonoids in the flesh and epidermis of red- and yellow-corm *P. ternata* to investigate the overall metabolic differences between the two color forms. A total of 283 flavonoids, including 7 chalcones, 21 flavanones, 12 flavanonols, 8 anthocyanins, 92 flavones, 54 flavonols, 35 flavonoid carbonosides, 21 flavanols, 17 tannins, 9 isoflavones, and 7 proanthocyanidins were detected using UPLC-MS/MS in our study (Appendix A).

Principal component analysis (PCA) was used to reveal the relative separation between the Yf, Ye, Rf, and Re samples (Figure 2A). The first principal component, accounting for 56.81% of the metabolic variance between the samples, separated the Rf and Yf samples from the Ye and Re samples. The second principal component, accounting for 22.43% of the metabolic variance, resulted in the distinct separation of the Rf samples from the Yf samples. Moreover, the third principal component, accounting for 8.8% of the metabolic variance, separated the Re samples from the Ye samples. These results indicated that the metabolic profiles were significantly different between the red- and yellow-corm *P. ternata* samples. The 88.04% variance among the flesh and epidermis samples of the two color forms of *P. ternata* indicated that the flavonoids exhibited different expression patterns in the two color forms. Cluster analysis of the metabolites in the epidermis and flesh of the two *P. ternata* lines showed that the Ye and Re samples clustered into one group, and the Yf and Rf samples clustered into another group, indicating that the flavonoids in the epidermis had a different expression pattern than those in the flesh of both color forms of *P. ternata* (Figure 2B).

To identify the differentially accumulated flavonoids in the four samples from the red and yellow *P. ternata* corms, we screened the metabolites with a cut-off of |Log2-ratio| ≥ 1 and VIP (variable importance in projection) ≥ 1. The pairwise comparison between Re and Ye showed that the contents of 84 flavonoids were higher, whereas the contents of 21 flavonoids and 5 tannins were lower in Re. In the Rf vs. Yf comparison, 48 flavonoids and 4 tannins were upregulated, and 11 flavonoids were downregulated in Rf. The number of differentially accumulated flavonoids and tannins in the Rf vs. Re comparison was 173 and 13, respectively, of which, 170 flavonoids and 13 tannins were upregulated and 3 flavonoids were downregulated in Re compared with Rf. Furthermore, a comparison between Yf and Ye showed that the contents of 155 flavonoids and 21 tannins were upregulated, and 4 flavonoids were downregulated in Ye compared with Yf (Appendix A). It can be seen from the above results that the content of flavonoid metabolites in the epidermis of *P. ternata* corms was higher than in the flesh. Moreover, the content of flavonoids in red *P. ternata* corms was higher than in yellow corms.

The identified differential metabolites from each pairwise combination were annotated using the Kyoto Encyclopedia of Genes and Genomes (KEGG) database. As a result, the differential metabolites in the comparison groups were assigned to the flavonoid biosynthesis (ko00941), anthocyanin biosynthesis (ko00942), isoflavonoid biosynthesis (ko00943), and flavone and flavonol biosynthesis (ko00944) pathways. Only 55 differential metabolites were enriched in the known biosynthesis pathways, including flavonoid biosynthesis (27, ko00941), anthocyanin biosynthesis (6, ko00942), isoflavonoid biosynthesis (5, ko00943), and flavone and flavonol biosynthesis (25, ko00944) (Figure 2C, Appendix A). In particular, the contents of all six anthocyanins—cyanidin-3-O-glucoside, pelargonidin-3-O-rutinoside, cyanidin-3-O-rutinoside, cyanidin-3-O-rutinoside-5-O-glucoside, pelargonidin-3,5-O-diglucoside, and cyanidin-3-O-rutinoside-5-O-glucoside—were highly elevated in Rf and Re compared with the levels in Yf and Ye. In the Ye vs. Re comparison, the contents of pelargonidin-3-O-rutinoside, cyanidin-3-O-rutinoside, cyanidin-3-O-rutinoside-5-O-glucoside, pelargonidin-3,5-O-diglucoside, and cyanidin-3-O-rutinoside-5-O-glucoside were much higher in Re than in Ye. In the Yf vs. Ye comparison, the contents of pelargonidin-3-O-glucoside, pelargonidin-3-O-rutinoside, cyanidin-3-O-rutinoside, and cyanidin-3-O-rutinoside-5-O-glucoside were higher in Ye than in Yf (Appendix A).

In addition, in order to comprehensively identify potential differential metabolite markers between the red and yellow corm forms of *P. ternata*, we established orthogonal projection models for latent structures discriminant analysis (OPLS-DA) for the Re vs. Ye and Rf vs. Yf comparisons (both Q2Y and R2 were all <0.05) (Appendix A). Among the DAMs in the Re vs. Ye and Rf vs. Yf comparisons, we identified 150 and 126 DAMs with VIP > 1.0, respectively. We found that there were 72 common DAMs in the Re vs. Ye and Rf vs. Yf comparisons, and 15 metabolites, including 1 chalcone (okanin), 1 tannin (1-O-galloyl-D-glucose), 4 flavonols (kaempferol-3-O-neohesperidoside-7-O-glucoside, quercetin-3-O-rutinoside (Rutin), quercetin-3-O-(2″-O-rhamnosyl) galactoside, and quercetin-7-O-rutinoside), 4 anthocyanins (cyanidin-3-O-rutinoside-5-O-glucoside, cyanidin-3-O-glucoside (kuromanin), peonidin-3-O-rutinoside, and cyanidin-3-O-rutinoside (keracyanin)), and 5 flavanonols (aromadendrin (dihydrokaempferol), taxifolin (dihydroquercetin), isookanin, butin, and naringenin (5,7,4′-trihydroxyflavanone)), with VIP values that ranked in the top 50 of the two groups of DAMs. Other than kaempferol-3-O-neohesperidoside-7-O-glucoside and 1-O-galloyl-D-glucose, in which the contents were higher in the Ye and Yf than in the Re and Rf groups, respectively, the contents of the other 13 compounds were all higher in the red corm samples compared with the yellow corm samples (Figure 2D). It is well known that aromadendrin (dihydrokaempferol), taxifolin (dihydroquercetin), and naringenin are important intermediates in anthocyanin biosynthesis, and that cyanidin-3-O-rutinoside-5-O-glucoside, cyanidin-3-O-glucoside, peonidin-3-O-rutinoside, and cyanidin-3-O-rutinoside are important anthocyanins because they impart the bright colors to flowers and other organs in many plants. Therefore, the color of the epidermis in red *P. ternata* corms must be due to the activity of the anthocyanin biosynthesis pathway (Appendix A).

### 2.2. Transcriptomic Analysis of the Flesh and Epidermis of Red and Yellow Corms of P. ternata

To investigate the molecular mechanism of epidermal color formation in *P. ternata* corms, a full-length transcriptome library was constructed from mixed tissue samples (root, corm, petiole, and leaf) from the red and yellow corm forms of *P. ternata*. The mRNAs in the Rf, Re, Yf, and Ye samples (three biological repeats of each) were subjected to nucleotide sequencing on an Illumina HiSeq 2500 instrument. A total of 78.61 Gb clean sequencing reads were obtained, and each sample yielded > 6 Gb data after the quality filtering process (Appendix A).

A total of 677,815 circular consensus sequencing (CCS) reads with an average length of 2108 bp were successfully generated using PacBio SMRT sequencing, of which, 543,296 were full-length non-chimeric (FLNC) reads. FLNC accuracy correction was performed using two strategies: (1) self-correction and (2) correction using next-generation Illumina sequencing data (125 base/150 base paired-end reads). Finally, 21,464 non-redundant transcripts with an average length of 2073 bp and N50 of 2427 bp were obtained (Appendix A). Among the 21,464 identified transcripts, 18,579 (87%) and 18,822 (88%) of the genes were functionally annotated in the public Nr and TrEMBL databases, respectively (Appendix A). Forty-seven percent of the genes could be assigned to gene ontology (GO) terms. GO enrichment analysis grouped the functions of these transcripts into the three GO categories, cellular component (CC), biological process (BP), and molecular function (MF) (Appendix A; Appendix A). In terms of BP ontology, protein phosphorylation (GO:0006468, 774 transcripts), protein binding (GO:0005515, 1223 transcripts), and protein kinase activity (GO:0004672, 774 transcripts) were highly enriched (Appendix A). The full-length transcriptome provided a high-quality transcript profile reference for *P. ternata* with many well annotated and full-length genes being represented.

### 2.3. Analysis of Differentially Expressed Genes (DEGs) in the Flesh and Epidermis of P. ternata Corms

Deseq2 [26,27] was used to analyze the differential gene expression between the Rf, Re, Yf, and Ye sample groups. Volcano plots were drawn to visually display the distribution of DEGs in each group. The red, green, and blue dots indicate the upregulated genes, the downregulated genes, and the genes in which the difference in expression was non-significant, respectively (Figure 3). Using fold-change ≥ 1 and FDR < 0.05 as the filter criteria for the DEGs, we identified 709 (including 492 upregulated genes and 217 downregulated genes), 1634 (728 upregulated genes and 906 downregulated genes), 704 (376 upregulated and 328 downregulated genes), and 2069 (1478 upregulated and 591 downregulated genes) DEGs in the Ye vs. Re, Yf vs. Rf, Rf vs. Re, and Yf vs. Ye comparisons, respectively. The DEGs from the Ye vs. Re comparison were significantly enriched in terms of biotin metabolism (ko00780, 31 DEGs), plant–pathogen interaction (ko04626, 96 DEGs), and the MAPK plant signaling pathway (ko04016, 53 DEGs). DEGs in the Yf vs. Rf group were significantly enriched in plant hormone signal transduction (ko04075, 29 DEGs), sphingolipid metabolism (ko00600, 12 DEGs), glycosphingolipid biosynthesis-ganglio series (ko00604, 6 DEGs), and other glycan degradation (ko00511, 8 DEGs). DEGs in the Rf vs. Re comparison were significantly enriched in terms of the biosynthesis of secondary metabolites (ko01110, 137 DEGs); phenylpropanoid biosynthesis (ko00940, 35 DEGs); starch and sucrose metabolism (ko00500, 37 DEGs); metabolic pathways (ko01100, 184 DEGs); cutin, suberine, and wax biosynthesis (ko00073, 9 DEGs); flavonoid biosynthesis (ko00941, 10 DEGs); phenylalanine metabolism(ko00360, 10 DEGs); fatty acid elongation (ko00062, 9 DEGs); stilbenoid, diarylheptanoid and gingerol biosynthesis (ko00945, 5 DEGs); zeatin biosynthesis (ko00908, 4 DEGs); and ubiquinone and other terpenoid-quinone biosynthesis pathways (ko00130, 10 DEGs). DEGs in the Yf vs. Ye comparison were significantly enriched in phenylpropanoid biosynthesis (ko00940, 41 DEGs); biotin metabolism (ko00780, 24 DEGs); the MAPK plant signaling pathway (ko04016, 66 DEGs); linoleic acid metabolism (ko00591, 14 DEGs); ubiquinone and other terpenoid-quinone biosynthesis pathways (ko00130, 22 DEGs); metabolic pathways (ko01100, 436 DEGs); the biosynthesis of unsaturated fatty acids (ko01040, 14 DEGs); fatty acid metabolism (ko01212, 27 DEGs); and sphingolipid metabolism (ko00600, 24 DEGs) (Figure 3).

As anthocyanins are products of the flavonoid biosynthesis pathway, we screened the DEGs related to flavonoid metabolism in each pairwise comparison group, and the genes with high expression levels (FPKM ≥ 10) in each group were retained. We identified 17 upregulated genes and 2 downregulated genes in the Rf vs. Re comparison, 8 upregulated genes and 11 downregulated genes in the Ye vs. Re comparison, 9 upregulated genes and 2 downregulated genes in the Yf vs. Rf comparison, and 19 upregulated genes and 5 downregulated genes in the Yf vs. Ye comparison. These genes are listed in Appendix A.

Anthocyanin biosynthesis in *P. ternata* occurs mainly in the epidermis of the corm. We identified seven DEGs, *PtCHS1*, *PtCHS2*, *PtCHI1*, *PtDFR5*, *PtANS*, *PtUPD-GT2*, and *PtUPD-GT3*, that were co-expressed in the Rf vs. Re and Yf vs. Ye comparisons. We hypothesized that these seven DEGs are the key factors involved in promoting the accumulation of anthocyanins in *P. ternata*. To understand the mechanism of the formation of the different epidermis colors in the red and yellow *P. ternata* corms, KEGG functional annotation was performed on the DEGs from the Ye vs. Re comparison. We identified 19 DEGs in the Ye vs. Re comparison: *PtF3H1*, *PtF3H2*, *PtF3H3*, *PtF3′H4*, *PtF3′H3*, *PtF3′5′H*, *PtDFR6*, *PtDFR2*, *PtUPD-GT1*, *PtUPD-GT11*, *PtUPD-GT12*, *PtUPD-GT16*, *PtUPD-GT15*, *PtUPD-GT14*, *PtMYB6*, *PtMYB7*, *PtMYB8*, *PtbHLH9*, and *PtbHLH10.* Furthermore, we noted that three of the 19 key genes, *PtF3′5′H*, *PtDFR2*, and *PtUPD-GT1*, were also identified as DEGs for anthocyanin biosynthesis in the Rf vs. Re comparison (*PtCHS1*, *PtCHS2*, *PtCHI1*, *PtF3′H1*, *PtF3′H2*, *PtF3′5′H*, *PtDFR1*, *PtDFR2*, *PtDFR3*, *PtDFR4*, *PtDFR5*, *PtANS*, *PtUPD-GT1*, *PtUPD-GT2*, *PtUPD-GT3*, *PtUPD-GT4*, *PtUPD-GT5*, *PtbHLH1*, and *PtbHLH2*). From this, we inferred that *PtF3′5′H*, *PtDFR2*, and *PtUPD-GT1* may be the genes responsible for the difference in epidermis color in the red and yellow corms of *P. ternata* (Figure 4).

The biosynthesis of anthocyanins is not only regulated by enzymes related to the anthocyanin metabolic pathway, but also by regulatory factors, such as MYB transcription factors, basic helix–loop–helix transcription factors (bHLH), and WD40 repeat proteins. A total of 12 *bHLHs* and 8 *MYBs* were identified in all of the comparison groups, including 2 *bHLHs* and 3 *MYBs* in the Ye vs. Re comparison, 4 *bHLHs* in the Yf vs. Rf comparison, 2 *bHLHs* in the Rf vs. Re comparison, and 6 *bHLHs* and 5 *MYBs* in the Yf vs. Ye comparison groups. *PtbHLH1* and *PtbHLH2* identified in the Rf vs. Re comparison may have a regulatory effect on the biosynthesis of anthocyanins in red *P. ternata* corms.

### 2.4. Association Analysis of Metabolic and Transcriptomic Data from Red and Yellow P. ternata Corms

To observe the correlation between the transcriptomic data and the metabolites in the anthocyanin biosynthesis pathway, we performed a correlation analysis on DEGs related to anthocyanin biosynthesis and DAMs involved in the anthocyanin metabolic pathway (*p* < 0.05, R > 0.8). The results showed that only *PtbHLH1* and *PtMYB6* had a significant correlation with dihydromyricetin, and all TFs had no significant correlation with anthocyanidins. A total of 13 DEGs, *PtCHS1*, *PtCHS2*, *PtCHI1*, *PtF3′H1*, *PtF3′H2*, *PtF3′H4*, *PtF3′5′H*, *PtDFR1*, *PtDFR2*, *PtUPD-GT1*, *PtUPD-GT2*, *PtUPD-GT3*, and *PtUPD-GT4*, showed a significant positive correlation with most DAMs involved in the anthocyanin metabolism pathways. Furthermore, *PtCHS1*, *PtCHS2*, *PtCHI1*, *PtF3′H1*, *PtF3′H2*, and *PtF3′H4* showed significant positive correlations with the enzymatic products of their encoded enzymes, naringenin chalcone, naringenin, and dihydroxyquercetin, respectively. Therefore, we speculated that these six genes play important roles in the biosynthesis of anthocyanins in *P. ternata* (Figure 5A).

To study the association between the transcriptomic data and the anthocyanidins present in the *P. ternata* corms, we performed a correlation analysis between the DEGs related to anthocyanin biosynthesis, and the contents of cyanidin and pelargonidin determined by HPLC (*p* < 0.05, R > 0.8). The results showed that five genes, *PtCHI1*, *PtDFR2*, *PtUPD-GT3*, *PtUPD-GT1*, and *PtUPD-GT2,* were significantly positively correlated with the contents of cyanidin and pelargonidin, whereas *PtF3H2* and *PtUPD-GT11* were significantly negatively correlated with the contents of cyanidin and pelargonidin (Figure 5B). Furthermore, we analyzed the correlations between cyanidin, pelargonidin, and the DAMs in the anthocyanin biosynthesis pathway (*p* < 0.05, R > 0.8). We found that nine metabolites, naringenin chalcone, naringenin, dihydrokaempferol, dihydroquercetin, cyanidin-3-O-glucoside, cyanidin-3-O-rutinoside, cyanidin-3-O-rutinoside-5-O-glucoside, pelargonidin-3-O-rutinoside, and peonidin-3-O-rutinoside, were significantly positively correlated with the cyanidin content, which is the main anthocyanin present in *Pinellia ternata* (Figure 5C).

On the basis of extensive investigations into the anthocyanin biosynthesis pathway in other plants [28,29,30], we constructed a possible route map for anthocyanin biosynthesis in *P. ternata* (Figure 6). According to the results of our correlation analysis, *PtCHI1*, *PtDFR2*, *PtUPD-GT1*, *PtUPD-GT2*, and *PtUPD-GT3* were identified as the key genes involved in the anthocyanin biosynthesis pathway in *P. ternata*. *PtCHI1*, which encodes chalcone isomerase is the second rate-limiting enzyme in plant flavonoid synthesis. Our results showed that the expression level of *PtCHI1* was markedly increased in red *P. ternata* corms, and that naringenin, the product of *PtCHI1*, was also highly enriched in the red corms. DFR is a key enzyme in anthocyanin biosynthesis that catalyzes the production of leucoanthocyanidins by using dihydrokaempferol, dihydroquercetin, and dihydromyricetin as substrates [31,32,33]. In this study, we found three DFR genes (*PtDFR2*, *PtDFR5,* and *PtDFR6*) that showed upregulated expression in red *P. ternata* corms. However, only the expression of *PtDFR2* was significantly positively correlated with the contents of cyanidin and pelargonidin. In addition, the substrate contents of dihydroquercetin and dihydrokaempferol were also positively correlated with the contents of cyanidin and pelargonidin, respectively. These results suggested that *PtDFR2* may be the key enzyme that catalyzes the formation of cyanidin and pelargonidin from dihydroquercetin and dihydrokaempferol, respectively. Glycosyltransferases (UPD-GTs) are the last key enzymes in anthocyanin biosynthesis that catalyze the conversion of unstable anthocyanidins into anthocyanins by glucosylation [34,35]. There were eight *PtUPD-GT* genes that were found to be highly expressed in red corms of *P. ternata*. Three of these, *PtUPD-GT1*, *PtUPD-GT2*, and *PtUPD-GT3*, may play a key role in the synthesis of cyanidin-3-O-rutinoside, cyanidin-3-O-rutinoside-5-O-glucoside, cyanidin-3-O-glucoside, pelargonidin-3,5-O-diglucoside, and pelargonidin-3-O-rutinoside.

### 2.5. qRT-PCR Validation

To verify the accuracy of the gene expression data obtained by RNA-seq, 10 DEGs related to anthocyanin biosynthesis, including the five candidate genes, were selected for qRT-PCR expression analysis. The expression level of the 10 DEGs shown by qRT-PCR was highly consistent with that shown by RNA-seq (Figure 7), which indicated that the transcriptome-based analysis of DEGs was highly reliable.

## 3. Discussion

The color of plants is an important characteristic of many plants. The different colors of plants are caused by the differences in the content of certain metabolites [36,37,38,39,40,41,42]. Therefore, the differences in apparent color can be considered as the expression of physiological and biochemical differences in plants. This not only gives plants better resistance to biological and abiotic stresses, but also changes the flavors of fruits, and even improves the contents of active ingredients in medicinal plants. Changes in the apparent colors of plants are generally related to the varied contents of carotenoids, anthocyanins, and diterpenoids, with anthocyanins being the main factor. Whether in the aboveground or underground parts of plants, changes in the anthocyanin content will lead to significant changes in appearance, which can be beneficial to the plant. For example, anthocyanins can enhance the ability of tea (*Camellia sinensis*) plants to resist bacteria and fungi [39]. Potato [40] and onion [41] with higher anthocyanin contents have stronger antioxidant capacities. *Panax notoginseng* [42] which has a purple color due to its high anthocyanin content, is considered to have a greater medicinal value.

However, as an important medicinal plant, *P. ternata* has been the subject of many studies. In previous studies, researchers often focused on the growth-influencing factors [43,44], chemical constituents [16,17], and pharmacological effects [45,46] of *P. ternata*. At present, there has been no research undertaken on the color difference in corms, which is undoubtedly a major deficiency in the breeding of *P. ternata*. Based on the resource survey of *P. ternata*, our team found a naturally occurring variant of *P. ternata* in which the corms have a red epidermis. We found that the total flavonoid content of red *P. ternata* corms was nearly 1.6 times that of yellow *P. ternata* corms, and red *P. ternata* were more resistant to environmental stresses than yellow *P. ternata* in the field.

In this study, we combined metabolomics and transcriptomic analysis to investigate the molecular mechanisms behind the difference in epidermal color between the two forms of *P. ternate*. As is shown in Figure 3, the DEGs in the Ye vs. Re comparison were significantly enriched in the plant–pathogen interaction (ko04626) and MAPK plant signaling pathway (ko04016). Plants are sensitive to infection by bacteria or viruses, which is an important basis for plants to obtain biological resistance [47]. Studies have shown that the interaction between plants and pathogens promoted the evolution of a multi-layer immune system in plants to prevent or hinder colonization by pathogens [48]. Protein kinases are a class of enzymes that catalyze the phosphorylation of related proteins. Mitogen-activated protein kinases (MAPKs), which are members of the serine/threonine protein kinase family, are one of the most widely studied gene families. MAPKs play a positive role in regulating the ability of plants to resist the effects of stress caused by drought, extreme temperature, salt, and pathogen attack [49,50,51]. Therefore, we predicted that the red corms of *P. ternata* would have better biotic and abiotic resistance compared with the plants with yellow corms.

We found that the red color of *P. ternata* was due to the high concentration of anthocyanins in the epidermis of their corms. Furthermore, based on HPLC and metabolomics, we determined that the accumulation of cyanidin and pelargonidin led to the production of *P. ternata* epidermal red, and identified 15 potential markers. We then identified the genes in the anthocyanin biosynthetic pathway from the DEGS in the Yf vs. Ye and Rf vs. Re comparisons, and we found that seven genes were shared. *PtCHS1*, *PtCHS2*, *PtCHI1*, *PtDFR5*, *PtANS*, *PtUPD-GT*2, and *PtUPD-GT3* may therefore play important roles in the accumulation of anthocyanins in the epidermis of *P. ternata*. In the Ye vs. Re comparisons, *PtF3Hs* and *PtDFRs* were identified; the same results were also reported by Yin [52], suggesting that these genes may have great significance in the formation of epidermal red in *P. ternata*. Two TFs, *PtbHLH1* and *PtbHLH2*, were identified in the Rf vs. Re comparisons; they may be able to regulate the biosynthesis of anthocyanins in red *P. ternata*. A correlation analysis between the DEGs and DAMs of the anthocyanin biosynthesis pathway and the anthocyanin contents in *P. ternata* corms allowed us to identify key genes and metabolites related to anthocyanin content, which is more accurate at screening candidate genes. In addition, we used the full-length transcriptome data to correct the RNA-seq, which can obtain accurate gene and transcript expression quantification. Metabolites are the phenotypic expression of genes; naringenin chalcone is formed from the action of CHS on 4-coumaroyl-CoA and malonyl-CoA, and the content of naringenin chalcone is significantly positively correlated with the total anthocyanin content. Expression of both *PtCHS1* and *PtCHS2* was positively correlated with the anthocyanin content, but not significantly. This is thought to be the result of the joint action of the two genes, and the same situation occurred between *PtF3′H3*, *PtF3′H4*, and dihydroquercetin.

## 4. Materials and Methods

### 4.1. Plant Materials and Treatments

The experiments were carried out at the Laboratory of Traditional Chinese Medicine Resource Center, Hubei University of Traditional Chinese Medicine, from September to October 2021. Young *P. ternata* corms between 1.0 cm and 1.2 cm in size were selected and transplanted into culture pots (6 cm diameter and 6 cm height) filled with a mixture of matrix soil and perlite (matrix soil:perlite = 3:1). *P. ternata* corms of both color types were placed in the same growth chamber. The temperature inside the growth chamber was 25 °C day/21 °C night, the light intensity was 10,000 lux, and the photoperiod was 13 h light/11 h dark. On day 30 of growth, the epidermis and flesh of the two types of *P. ternata* corms were separated, immediately frozen in liquid nitrogen, and then stored in a freezer at −80 °C.

### 4.2. Determination of Total Flavonoid Content

Flavonoid concentrations were quantified using the method described by He [53] with a few modifications. Briefly, a calibrated linear function was established using rutin as a standard: y = 0.017x + 0.0061, R^2^ = 0.9991. Samples of 0.3 g were weighed, and the flavonoids in these samples were extracted with 20 mL of 50% ethanol in water by ultrasound for 30 min. A total of 5 mL of the extraction solution was mixed with 8 mL of 1.5% AlCl_3_ and 4 mL of acetic acid–sodium acetate buffer medium (pH 5.5). After 30 min, the absorption value of the mixed solution was measured using a spectrophotometer at a wavelength of 415 nm.

### 4.3. Determination of Total Anthocyanin Contents

The frozen samples were ground into a powder in liquid nitrogen using a mortar and pestle. To extract the anthocyanins, 200 mg of the sample powders were transferred to conical flasks, 10 mL of the extracting solvent (8.6 M CH_3_CH_2_OH, 3 M HCl) was added to each, and the total weights were recorded. The flasks were heated in a boiling water bath for 1 h following an ultrasonic treatment for 30 min. After heating, the flasks were allowed to cool, and the extracting solvent was added to each to make up the original weight. The conditions for HPLC were a C18 reverse phase column (250 mm × 4.6 mm × 5 μm); the mobile phase consisted of acetonitrile (containing 1% formic acid) and 1% formic acid in water with gradient elution (Appendix A), a flow rate of 0.8 mL min^−1^, and a column temperature of 30 °C; and the UV wavelength for detection was 530 nm. The sample volumes were 10 μL. Cyanidin and pelargonidin were dissolved in the extraction solvent at concentrations of 0.5, 1.0, 5.0, 10.0, and 20.0 mg/L as standards. The anthocyanin content in the samples was the sum of the contents of two anthocyanins, and the combined content was calculated by the mass fraction w. The unit was mg/kg: w = (ρ × V)/m, where ρ equals the mass concentration of anthocyanins in the solution to be measured, V equals 10 mL, and m is the sample weight.

### 4.4. Metabolite Extraction and Ultra-High-Performance Liquid Chromatography−Mass Spectrometry (UPLC-MS/MS) Analysis

The samples were lyophilized in a vacuum freeze-dryer (Scientz-100F; Ningbo Scientz Biotechnology Co., Ningbo, China) and then crushed using a mixer mill (MM 400, Retsch) with a zirconia bead for 1.5 min at 30 Hz. Samples (100 mg) of the lyophilized powders were suspended in a 1.2 mL 70% methanol solution and vortex mixed for 30 s every 30 min, a total of six times. Afterward, the samples were refrigerated at 4 °C overnight. The samples were then centrifuged at 12,000× *g* rpm for 10 min, and the cleared extracts were filtered through membranes (SCAA-104, 0.22 μm pore size) prior to UPLC-MS/MS analysis. The sample extracts were analyzed using a UPLC-ESI-MS/MS system (UPLC, Shimadzu Nexera X2; MS, Applied Biosystems 4500 Q TRAP). The analytical conditions were as follows for UPLC: column, Agilent SB-C18 (1.8 µm, 2.1 mm × 100 mm); the mobile phase consisted of solvent A, 0.1% formic acid in water; and solvent B, acetonitrile containing 0.1% formic acid. Sample measurements were performed with a gradient program starting with 95% A, 5% B. Within 9 min, a linear gradient going to 5% A, 95% B was programmed, and a composition of 5% A, 95% B was maintained for 1 min. Subsequently, a composition of 95% A and 5.0% B was adjusted within 1.1 min and maintained for 2.9 min. The flow rate was 0.35 mL per minute, the column oven was set to 40 °C, and the injection volume was 4 μL. The effluent was alternatively connected to an ESI-triple quadrupole–linear ion trap (QTRAP)-MS. The linear ion trap (LIT) and triple quadrupole (QQQ) scans were acquired on a triple quadrupole–linear ion trap mass spectrometer (Q TRAP), AB4500 Q TRAP UPLC/MS/MS System, equipped with an ESI turbo ion spray interface, operating in a positive and negative ion mode and controlled by Analyst 1.6.3 software (AB Sciex). The ESI source operation parameters were as follows: an ion source, turbo spray; source temperature 550 °C; ion spray voltage (IS) 5500 V (positive ion mode)/−4500 V (negative ion mode); the ion source gas I (GSI), gas II (GSII), curtain gas (CUR) were set at 50, 60, and 25.0 psi, respectively; the collision-activated dissociation (CAD) was high. Instrument tuning and mass calibration were performed with 10 and 100 μmol/L of polypropylene glycol solutions in QQQ and LIT modes, respectively. QQQ scans were acquired as MRM (multiple reaction monitoring) experiments with the collision gas (nitrogen) set to medium. Declustering potential (DP) and collision energy (CE) for individual MRM transitions were carried out with further DP and CE optimization. A specific set of MRM transitions were monitored for each period based on the metabolites eluted within this period.

### 4.5. cDNA Library Construction and Sequencing

Total RNAs from the two types of *P. ternata* corms were extracted using the RNAprep Pure Plant Plus Kit (TIANGEN Biotech (Beijing) Co., Ltd., Beijing, China). RNA degradation and contamination were monitored visually on 1% agarose gels. RNA purity was determined using the NanoPhotometer^®^ spectrophotometer (Implen Inc., Munich, Germany), and RNA concentration was measured using the Qubit^®^ RNA Assay Kit in the Qubit 2.0 Fluorometer (Thermo Fisher Scientific Inc., San Jose, CA, USA). RNA integrity was assessed using the RNA Nano 6000 Assay Kit in the Bioanalyzer 2100 system (Agilent Technologies, Co., Ltd., Santa Clara, CA, USA). A total amount of 1 µg RNA per sample was used as the input material for the RNA sample preparations. Sequencing libraries were generated using the NEBNext^®^ UltraTM RNA Library Prep Kit for Illumina^®^ (New England Biolabs, Ipswich, MA, USA) following the manufacturer’s recommendations, and index codes were added so that the sequences could be attributed to each sample. Briefly, mRNA was purified from the total RNA using poly-T oligo-attached magnetic beads. RNA was fragmented using divalent cations at elevated temperature in NEBNext First Strand Synthesis Reaction Buffer (5×). First-strand cDNA was synthesized using random hexamer primers and M-MuLV reverse transcriptase (RNase H−). Second-strand cDNA synthesis was subsequently performed using DNA polymerase I and RNase H. Remaining overhangs were converted into blunt ends via exonuclease/polymerase activities. After adenylation of the 3′ ends of the cDNA fragments, the NEBNext Adaptor with a hairpin loop structure was ligated to prepare for hybridization. In order to select cDNA fragments of the preferred length (250–300 bp), the library fragments were purified using the AMPure XP system (Beckman Coulter, Beverly, CA, USA). Following size selection, 3 µL of the USER Enzyme (New England Biolabs, Ipswich, MA, USA) was used with the size-selected, adaptor-ligated cDNA at 37 °C for 15 min, followed by 5 min at 95 °C, prior to PCR amplification. PCR was performed using Phusion High-Fidelity DNA polymerase (NEB), Universal PCR primers, and Index (X) Primer. Finally, the PCR products were purified using the AMPure XP system and the library quality was assessed with the Agilent Bioanalyzer 2100 system. The clustering of the index-coded samples was performed on a cBot Cluster Generation System using TruSeq PE Cluster Kit v3-cBot-HS (Illumina, Inc., San Diego, CA, USA) as directed by the manufacturer. After cluster generation, the library preparations were sequenced on an Illumina Hiseq platform to generate 125 base/150 base paired-end reads. For the construction and sequencing of the full-length transcript library, equal quantities of total RNA from six samples (roots, corms, leaves of red and yellow corms *P. ternata*) were mixed together to prepare SMRT libraries [54]. Subsequently, the libraries were sequenced using a Pacific Biosciences RS sequencing instrument and the sequence data were processed using SMRT link software (V5.0, Pacific Biosciences of California, Inc., San Francisco, CA, USA). Circular consensus sequences (CCSs) were generated from subreads BAM files, and obtain unigenes based on reported methods [55]. All transcriptome data sets in this study have been uploaded to the NCBI SRA database (PRJNA893095).

### 4.6. Differentially Expressed Gene Analysis

Gene expression levels were estimated by RSEM, and the FPKM value for each gene was calculated based on the gene length. FPKM (fragments per kilobase of transcript per million fragments mapped) is currently the most commonly used method to estimate gene expression levels from RNA-seq data. For our samples, DESeq2 v1.22.1 was used to analyze the differential gene expression between the two groups. The *p*-values were corrected using the Benjamin and Hochberg method. The corrected *p*-values and |log2 fold-change| were used as the threshold for determining significant differences in gene expression.

### 4.7. Correlation Analysis

Cloud platform analysis tools, “https://cloud.metware.cn/#/tools/tool-List (accessed on 27 April 2023)”, were subjected to correlation analysis of “2.4” (*p*-value < 0.05, R > 0.8).

### 4.8. Confirmation of DEGs by Quantitative Real-Time PCR

The transcription levels of 10 DEGs related to anthocyanin biosynthesis, including the five candidate genes, were analyzed using qRT-PCR (gene-specific primers are provided in Appendix A). The total RNA of *P. ternata* corms was isolated using TRIzol (Tiangen Bio Co., Ltd., Beijing, China), and cDNA was synthesized using MLV reverse transcriptase (Promega Corporation, Madison, WI, USA). RT–qPCR was conducted using RealUniversal Color PreMix (SYBR Green) according to the manufacturer’s instructions (Tiangen, Beijing, China). The *P. ternata* 18S gene was used as the endogenous reference gene. The relative gene expression of each gene was determined using the 2^−Δt^ method, E*_g_* = 2^−ΔCp^ = 2^−(G−S)^. E_g_, the relative expression level of genes; G, Cp values of genes; S, Cp values of 18S gene.

## 5. Conclusions

In summary, we believe that the red color on the epidermis of *P. ternata* is an important epigenetic characteristic of its significantly increased total flavonoid content. Based on transcriptome analysis and metabolomics, we plotted the pathway of anthocyanin biosynthesis in corms of *P. ternata* and analyzed the molecular mechanism behind the formation of a red epidermis in *P. ternata* corms. The key genes for the enzymes involved in anthocyanin biosynthesis in red *P. ternata* corms were identified, and 15 potential markers for distinguishing red and yellow *P. ternata* corms were screened. Our results provide a theoretical basis for the molecular breeding of *P. ternata* and for the rapid identification of lines with red and yellow corms. Based on the interactions between genes and metabolites in the anthocyanin biosynthesis pathway, we conclude that there are five key genes, *PtCHI1*, *PtDFR2*, *PtUPD-GT1*, *PtUPD-GT2*, and *PtUPD-GT3*, in the formation of the red epidermis in *P. ternata* corms. Furthermore, naringenin chalcone, naringenin, dihydrokaempferol, and dihydroquercetin are important intermediates for anthocyanin accumulation in the epidermis of *P. ternata* corms.

## Figures and Tables

**Figure 1 ijms-24-07990-f001:**
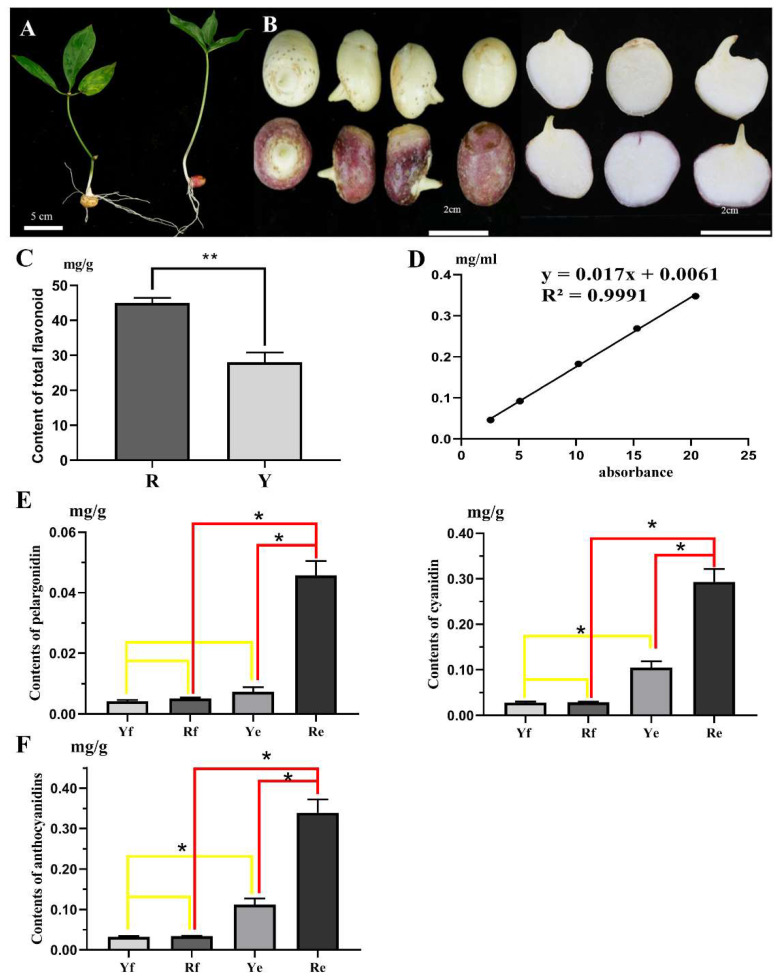
The phenotypes and anthocyanidin contents of two different colored *P. ternata*. (**A**) Phenotypes of yellow-corm and red-corm *P. ternata* plants. (**B**) The characteristics of red-corm *P. ternata* and yellow-corm *P. ternata*; the upper row is yellow-corm *P. ternata*, and the lower row is red-corm *P. ternata*. (**C**) Detection of the total flavonoid content of *P. ternata* samples. Values represent the means ± SD from three biological replicates (** *p* < 0.01, Student’s *t*-test). (**D**) A calibrated linear function was established using rutin as a standard; y, the absorbance values of samples; x, the concentration of total flavonoids of sample (mg/mL). (**E**) Detection of anthocyanidin content in *P. ternata* samples. Values represent the means ± SD from three biological replicates (* *p* < 0.05, Student’s *t*-test). (**F**) Total anthocyanidin contents in Yf, Rf, Ye, and Re samples.

**Figure 2 ijms-24-07990-f002:**
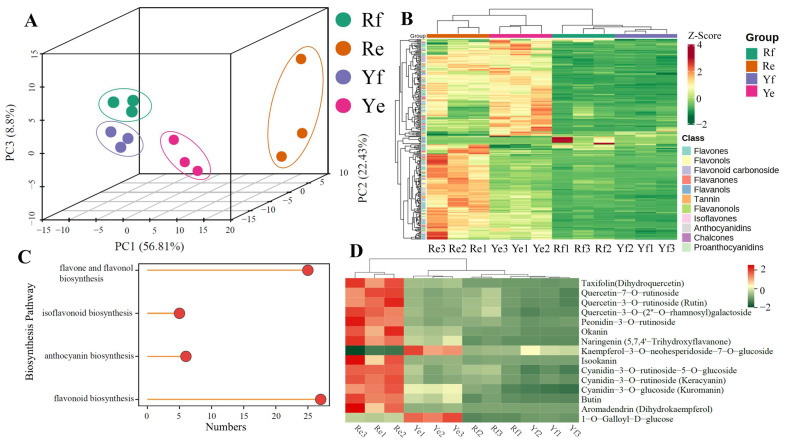
Metabolomic data analysis of two *P. ternata lines*. (**A**) Principal component analysis (PCA) score plots for all samples. (**B**) Cluster heat map of the 251 DAMs in the epidermis and flesh of two lines of *P. ternata*. (**C**) Enrichment of 55 differential metabolites and metabolic pathway differential metabolites. (**D**) Heat map of the 15 potential differential makers.

**Figure 3 ijms-24-07990-f003:**
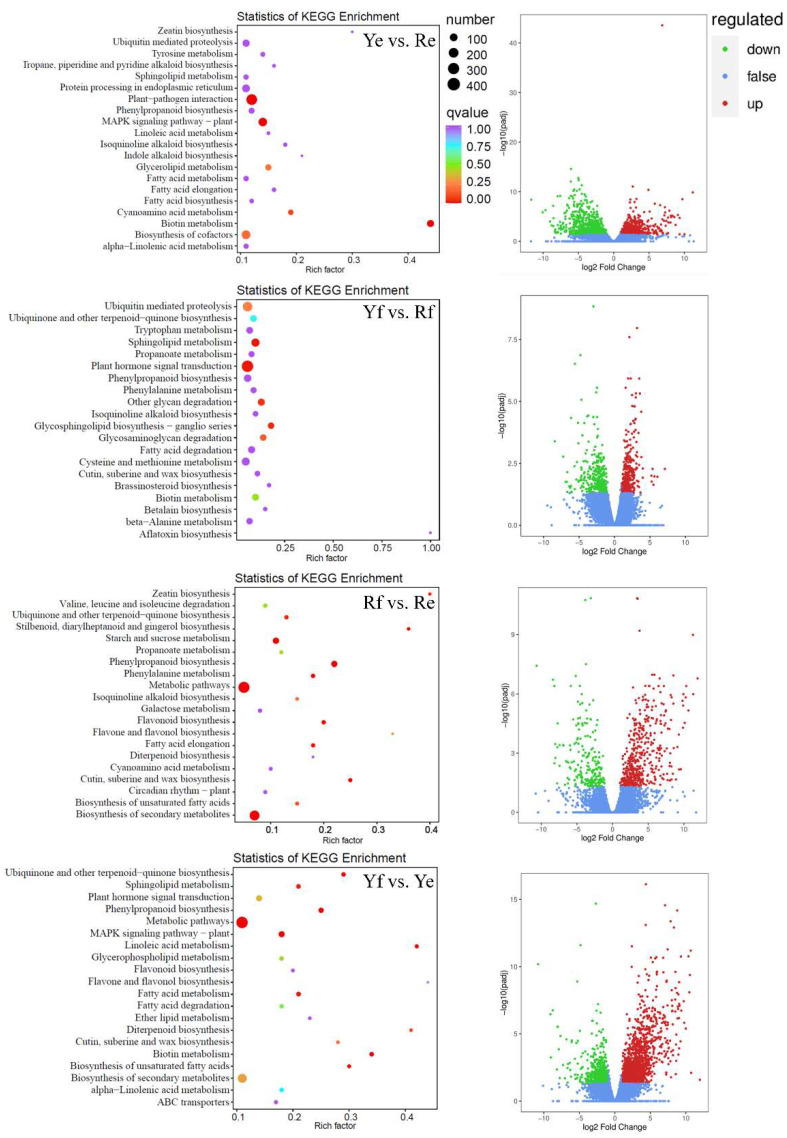
Volcano plots and KEGG enrichment analysis of differentially expressed genes (DEGs) in Ye vs Re, Yf vs Rf, Rf vs Re, and Yf vs Ye groups, respectively.

**Figure 4 ijms-24-07990-f004:**
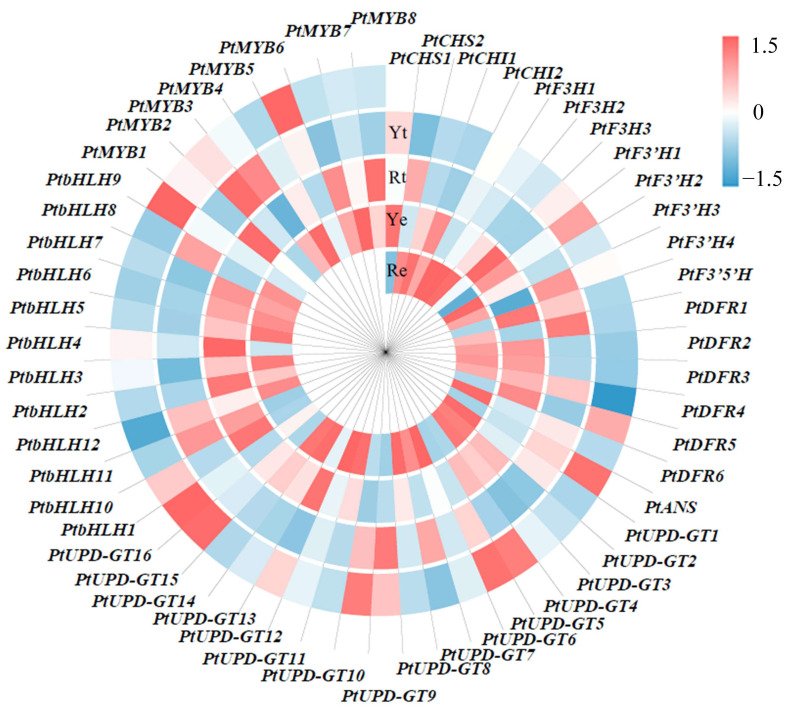
Heat map showing the expression of DEGs in the anthocyanin biosynthesis pathway in Re, Ye, Rf, and Rf.

**Figure 5 ijms-24-07990-f005:**
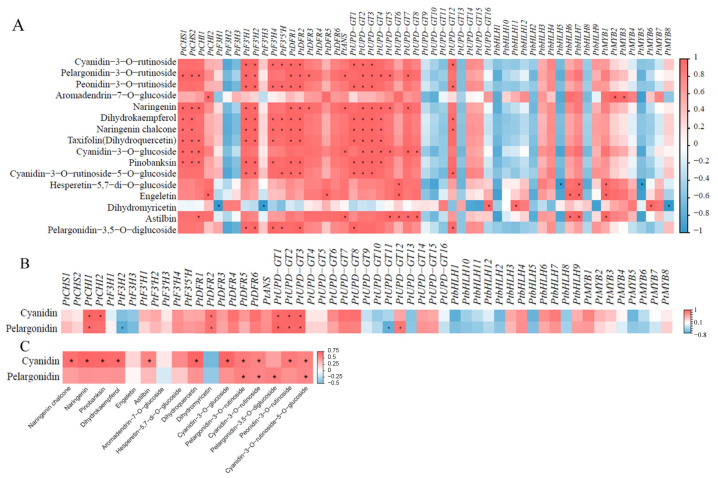
Correlation analysis of DEGs and DAMs related to anthocyanin biosynthesis. (**A**) Correlation analysis between DEGs related to anthocyanin biosynthesis and DAMs involved in anthocyanin metabolism. (**B**) Correlation analysis between DEGs related to anthocyanin biosynthesis and the contents of cyanidin and pelargonidin determined by HPLC. (**C**) Correlation analysis between DAMs involved in anthocyanin metabolism and the contents of cyanidin and pelargonidin determined by HPLC. *, significant correlation, *p* < 0.05, R > 0.8).

**Figure 6 ijms-24-07990-f006:**
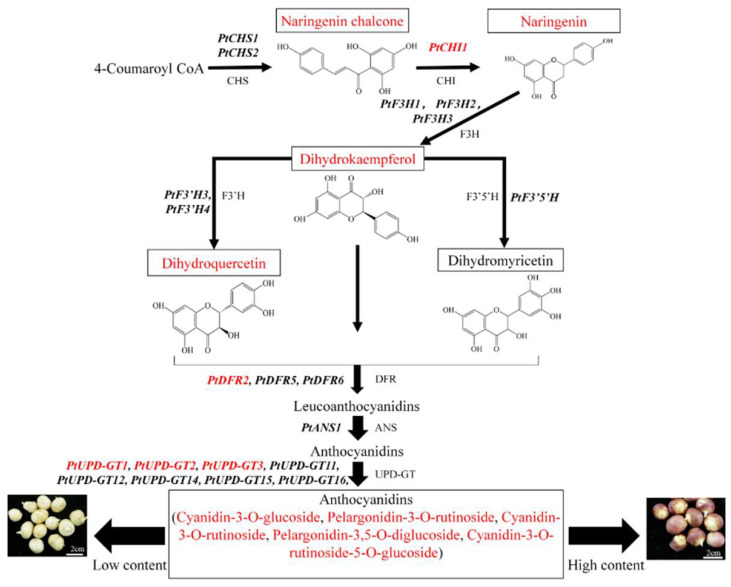
Diagram depicting the pathway of anthocyanin biosynthesis in *P. ternata*. The genes and metabolites in red are the key agencies leading to the color formation in corms of *P. ternata*.

**Figure 7 ijms-24-07990-f007:**
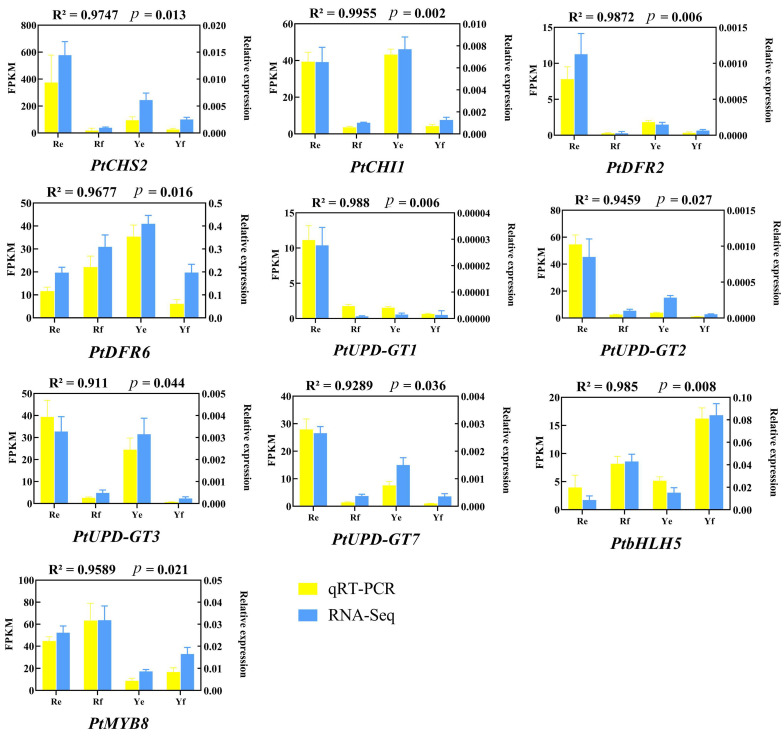
qRT-PCR confirmation of DEGs identified by transcriptome analysis.

## Data Availability

All transcriptome data sets in this study have been uploaded to the NCBI SRA database (PRJNA893095).

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
