# Peer review of "Integrative Analysis of Metabolomic and Transcriptomic Data Reveals the Mechanism of Color Formation in Corms of Pinellia ternata"

_ijms, 2023, doi:10.3390/ijms24097990_

Round 1

Reviewer 1 Report

In this study, Xu et al. conducted metabolome and transcriptome analyses for the molecular mechanism for the color formation in red- and yellow-corms of Pinellia ternate. The pathway analysis of anthocyanin biosynthesis may be the important part in this study. However, there are some problems should be solved in the present manuscript.

1. Tense. The Result section should be described in simple past tense, including the result description in the Abstract.

2. The Introduction section is complex. The authors should point out the aims of studying the corm color formation in P. ternate.

3. L46. The variant of one species should have its own Latin name, as ‘var.’. In this study, I think there are two germplasms with different corm colors.

4. L55. Salvia miltiorrhiza should be italic.

5. L106-108. It’s part of the Introduction, not the Result.

6. L115-116. Unclear.

7. L234-264. It’s interesting that there were more DEGs identified in the Yf vs Rf and Yf vs Ye comparisons, of which the KEGG annotated into some pathways related to anthocyanin. And more, how many DEGs for each pathway?

8. L268-274. List the DEGs related to anthocyanin biosynthesis.

9. Figure 4. Is there no DEGs related to WD40 or MT?

10. L294 and 378. There is no correlation analysis between DAMs and DEGs.

11. L301. What’s the Pearson coefficient as the negative correlation?

12. Figure 5. The figure legend is unclear.

13. M&M section. Add the description of correlation analysis. No qRT-PCR verification?

14. The authors could add some analysis of TFs to identify the key MYB members during corm coloration in P. ternate, like the similar analysis in Rosa rugosa (doi: 10.3389/fpls.2022.1021521).

15. The English writing should be improved.

Author Response

Dear editors and reviewers:

We are grateful for getting such professional suggestions and helpful criticism to revise our manuscript. We carefully considered every comment, and revised the manuscript accordingly. The tracing version indicates the changes in the mark-up manuscript for review only, and we also submit a clean manuscript without changes marked. We hope it is acceptable for publication. The following are our detailed responses (point-to-point answers in red font) to the editor’s and the reviewers’ comments.

We look forward to hearing from you soon.

Yours sincerely,

Prof. Dahui Liu & Dr. Yuhuan Miao

Hubei University of Chinese Medicine

-------------------------------------------------------------------------------

Reviewer 2 Report

The manuscript by Rong et al. uses biochemical, bioinformatic, and metabolic approaches to investigate the mechanism of color formation in P. ternate corms. The authors have generated a solid amount of data and the conclusions are justified by the results. However, some experiments and several details need to be clarified in the text or legends.

1. In Results part.

Line 108-109: “In our study, the anthocyanidin contents in the Yf, Ye, Rf, and Re samples were measured by HPLC”, typo “anthocyanidin” and it would be better to show the HPLC profile.

Line 301-302: “while PtF3H2 and PtUPD-GT11 were significantly negatively correlated with the contents of cyanidin and pelargonidin”, what could be the explanation for this result? Modify flux?

2. In Discussion part.

Why multiple genes for enzymatic steps were upregulated? Have you check any transcription factors? How are the bHLHs, MYBs, and WD40s?

3. In Materials and Methods part.

Line 398-402: “Research manuscripts reporting large datasets that are deposited in a publicly available database should specify where the data have been deposited and provide the relevant accession numbers. If the accession numbers have not yet been obtained at the time of submission, please state that they will be provided during review. They must be provided prior to publication”, delete?

Author Response

(The authors gave the same response as above.)

Reviewer 3 Report

In this paper by Xu and coworkers, the authors carry out a transcriptional and metabolomic analysis of Pinellia ternata in order to evaluate the molecular determinants for the corms  colour. As a general comment I would suggest to authors to generally improve the quality of the ms, for example taking care of the details, such as indicating all the species names in italic. Moreover I found several typos along the text, here reported some:

L312 typo in genes 

L499 typo in Benjamin

L505 typo behind

Then, please check the text from line 398 to 402, I think that this shoul not be there. 

Moving to the "scietific part", in general the quality of the work is good and should be published, however some issue need to be fixed:

major issues:

The intoduction should be revised adding appropriate citation, indeed from L68 to L83 there is no reference at all.

L496-497: what do the authors means for "samples without biological replicates"? In general transcriptomic experiments should be carried out both techical and biological replicates. I think this should be done also in this case.

As a result of the transcriptomic analyses the authors found 5 cadidate genes that are probably involved in the epidermis colour formation. I think in this case a qPCR validation for these 5 genes is needed.

Very recently a paper by Yin and coworker (Yin et al., 2023. Protoplasma - https://doi.org/10.1007/s00709-023-01845-7) applied a very similar approach to investigate the color formation of the tubers of Pinellia. Differences and similitudes between the two works are worth to be discussed by the authors. 

minor comment:

I would sugest to move the KEGG statistics legend in figure 3 on the right part of the figure, it would be easier to understand.

Author Response

(The authors gave the same response as above.)

Reviewer 4 Report

The present work unravels the difference in corm color in Pinelia ternata, a Chinese plant widely used in traditional medicine. The authors explained the differences occurring in corm color by the difference in anthocyanin content. They also identified several candidates by analyzing the metabolome and the transcriptome, thus contributing to the understanding of the color government in this plant species.

The manuscript comprises some minor mistakes that need to be corrected ( Please see the pdf file). Besides, the quality of the figures is poor and some are too small to read the content. I would recommend increasing the quality and the size of the figures.

Author Response

(The authors gave the same response as above.)

Round 2

Reviewer 1 Report

In the present manuscript, there are some places that confuse me.

For the Introduction section, what the main compound of Pinellia ternate for its medicine application, and what’s the relationship between its medicine value and corm color? It’s hard to understand the purpose of this study.

Transcription factor identification and expression analysis is an important part for the correlation analysis between metabolome and transcriptome analysis. The authors should add the related analysis and identify the key TFs for the color formation. There were no TF analysis in Abstract, Discussion, and Conclusion sections.

I haven’t seen much work of English language polishing, especially in the Discussion section. The first sentence: Color is as important characteristics in many plants. Color of plants?

L24-25: The GT1/2/3 were also structural genes.

L312/316: Please ensure the singular and plural forms.

L377: “…showed…”.

Author Response

Dear editors and reviewers:

Thank you again for your email and also the reviewers’ comments on our manuscript “Integrative Analysis of Metabolomic And Transcriptomic Data Reveals The Mechanism of Color Formation in Corms of Pinellia ternata”. We carefully considered every comment, and revised the text accordingly. The language of the manuscript was polished by the editing services recommend by the editor. The tracing version indicates the changes in the mark-up manuscript for review only, and we also submit a clean manuscript without changes marked. We would like to resubmit this revised manuscript to International Journal of Morlecular Science and hope it is acceptable for publication. The following are our detailed responses (point-to-point answers in red font) to the editor’s and the reviewers’ comments.

We look forward to hearing from you soon.

Yours sincerely,

Yuhuan Miao

Corresponding Author (On behalf of the Authors)

Reviewer 3 Report

In this second version of the ms the authors made significative effort to fix all the issues I raised, I really appreciate this. I think the general quality of the ms is improved with respect to the previous version. I still have some minor points that, in my opinion, need to be fixed before the publication:

L41-:L44: I don't think it is appropriate to state the efficiency of the traditional medicine as tratment against COVID-19. If peer-reviewed studies (that I'm not aware)  exist it is necessary to cite them, otherwise I would suggest to avoid claiming "In the present battle against COVID-19, both Qingfei Paidu Decoction and Huashi Baidu Granule, which have significant curative effects on COVID-19 infection, contain P. ternata." 

Figure 3:  I still find the legend of the figure a bit-confusing. I think it is better to move "number 100 200 300 400" and qvalue to the right side of the plot "Statistics of KEGG Enrichment" 

Figure 7: I really like this new figure. In this case I would suggest to add in the methods how the authors calculated the fold change in the qRT-PCR experiment.

Author Response

(The authors gave the same response as above.)

Round 3

Reviewer 1 Report

There are still some sentences that have not been changed. L396: showed, not shown. PLEASE CHECK ALL.